# Hormonal Contraceptives, Female Sexual Dysfunction, and Managing Strategies: A Review

**DOI:** 10.3390/jcm8060908

**Published:** 2019-06-25

**Authors:** Nerea M. Casado-Espada, Rubén de Alarcón, Javier I. de la Iglesia-Larrad, Berta Bote-Bonaechea, Ángel L. Montejo

**Affiliations:** 1Psychiatry Service, Institute of Biomedical Research of Salamanca (IBSAL), University Clinical Hospital of Salamanca, Paseo San Vicente, SN 37007 Salamanca, Spain; nmcasado91@gmail.com (N.M.C.-E.); ruperghost@gmail.com (R.d.A.); javidelaiglesia.jdli@gmail.com (J.I.d.l.I.-L.); bertabot@yahoo.es (B.B.-B.); 2Nursing School E.U.E.F., University of Salamanca, Av. Donantes de Sangre SN 37007 Salamanca, Spain

**Keywords:** female sexual dysfunction, hormonal contraceptive, libido, desire, sex life, orgasm, vaginal ring, depot medroxyprogesterone acetate

## Abstract

In recent decades, hormonal contraceptives (HC) has made a difference in the control of female fertility, taking an unequivocal role in improving contraceptive efficacy. Some side effects of hormonal treatments have been carefully studied. However, the influence of these drugs on female sexual functioning is not so clear, although variations in the plasma levels of sexual hormones could be associated with sexual dysfunction. Permanent hormonal modifications, during menopause or caused by some endocrine pathologies, could be directly related to sexual dysfunction in some cases but not in all of them. HC use seems to be responsible for a decrease of circulating androgen, estradiol, and progesterone levels, as well as for the inhibition of oxytocin functioning. Hormonal contraceptive use could alter women’s pair-bonding behavior, reduce neural response to the expectation of erotic stimuli, and increase sexual jealousy. There are contradictory results from different studies regarding the association between sexual dysfunction and hormonal contraceptives, so it could be firmly said that additional research is needed. When contraceptive-related female sexual dysfunction is suspected, the recommended therapy is the discontinuation of contraceptives with consideration of an alternative method, such as levonorgestrel-releasing intrauterine systems, copper intrauterine contraceptives, etonogestrel implants, the permanent sterilization of either partner (when future fertility is not desired), or a contraceptive ring.

## 1. Introduction

In recent decades, hormonal contraception (HC) has made a difference in the control of female fertility, taking an unequivocal role in improving contraceptive efficacy. Moreover, there are numerous studies that state that the use of hormonal contraceptives is very prevalent in the female population of childbearing age [1,2,3,4,5,6,7,8]. In a study carried out by Hall et al. in 2012, it was estimated that 63% of women of reproductive age worldwide who were married or in a relationship were using some type of contraception, with the contraceptive pill as the third most commonly used method (9% of women aged 15–19 years) [3,9]. Combined oral contraception seems to be the most popular form of reversible contraception in Europe and the United States [7,8].

The popularity and widespread use of hormonal contraceptives is partly due to their benefits, such as: (1) Being a highly effective and reversible form of contraception; (2) the woman has control over this method of contraception; (3) the failure rate is less than 1%; and (4) they have a well-established safety profile [1].

However, the use of hormonal contraceptives is relatively recent: In 1956, an oral contraceptive pill (mestranol in combination with norethynodrel) was used for the first time in a clinical trial; a year later, in 1957, the formulation of 150 μg mestranol and 10 mg norethynodrel received approval for the treatment of “female disorders” (menstrual irregularities, etc.) [1]. It was three years later, in 1960, when the Food and Drug Administration (FDA) approved the use of the pill as a contraceptive, containing 75 mestranol and 5 mg norethynodrel [1,10]. At the beginning, oral contraceptives were available only to married women, and, in 1972, the pill also began to be available for single women in all states [1]. Since the approval of the use of the pill in 1960, it has undergone many evolutions in dosage, hormone type, and regimen. It has been used by more than 100 million women worldwide and has the widest geographic distribution of any method of contraception [10].

The use of hormonal contraceptives is widespread, with a significant percentage of healthy population among its users. Some of its side effects are well known, such as the increased prothrombotic and cardiovascular risk (estrogen dependent) [10]. On the other hand, non-contraceptive benefits of hormonal contraceptives, such as as cycle regulation with predictable withdrawal bleeds, decreased menstrual flow, and decreased anemia, have been widely documented [10]. However, the influence of these drugs on female sexual function is not as clear, although it is mentioned in the technical prospects of the contraceptive pills. Additionally, there are very few controlled studies in this field.

Conversely, despite the widespread use of contraceptives in the general population, there are many other drugs that have been widely studied and associated with frequent iatrogenic sexual dysfunction. Antihypertensive drugs, diuretics, and beta-blockers seem to exert a detrimental impact on sexual function [11], as do antipsychotics [12,13,14], antidepressants [12,13,15], and others. In addition, there are endocrine disorders that are also associated with alterations in sexual function, such as diabetes [16], obesity, and metabolic syndrome [17]. On top of this, sexual dysfunction is a possible symptom associated with other hormonal alterations such as those that take place during menopause [18] or postpartum [19]. There are differences regarding which aspects of sexual function were most affected by menopause. The Massachusetts Women’s Health Study, the Melbourne Women’s Midlife Health project, the Penn Ovarian Aging Study and the Study of Women’s Health Across the Nation (SWAN) are some of the pieces of research that were carried out in this regard. Notably, three out of four of these studies noted declines in sexual desire during the menopause transition [18].

In this review, first of all, detailed information has been included about the hormonal contraceptive methods, focusing on the type of administration, hormonal composition, mechanism of action, and expected effects on hormonal function in women. Second, an approximation is given to the concept and significance of sexual dysfunction, in addition to its prevalence in the female population. These first sections have the objective of contextualizing and favoring the understanding of the next ones; the main aim of this study is to clarify whether there is evidence of the effect of hormonal contraceptives on female sexual function. In this review, we attempt to provide a summary of the existing data about the impact of hormonal contraceptives on sexuality. We differentiate between studies that claim that there are no effects of hormonal contraceptives on sexual function and others that defend that there are. Within the latter, we differentiate between those that show positive and negative effects on female sexual function. Likewise, in this review, some treatment options are proposed according to the studies reviewed.

This review provides a compilation of the existing evidence about the relationship between female sexual function and hormonal contraceptives, in addition to the existing therapeutic management strategies. This is the first review that includes a summary table, which allows the clinician to access to the most relevant information at a glance. Likewise, it is the only study that proposes a therapeutic algorithm for the management of hormonal contraceptives-related sexual dysfunction.

## 2. Materials and Methods

The aim of this review is developing, assimilating, and synthesizing the existing evidence about the influence of hormonal contraception on female sexual function. In addition, we intended to identify gaps in knowledge in this field in order to design new studies that may fill those gaps in the future. Our review focuses on the use of hormonal contraceptives in women of childbearing age and on the influence of these drugs on female sexual function [1,2]. In addition, the study reviews the differences in the influence of the HCs on female sexual function (FSF) according to the hormonal composition and the mechanism of action of the different HCs in order to determine which one has the lowest profile of secondary effects in the sexual area. On the other hand, to our knowledge, this is the latest effort to offer an overview of the recommended strategies in cases in which the use of HCs is associated with sexual dysfunction.To achieve this purpose, we performed a scoping review following PRISMA guidelines (Preferred Reporting Items for Systematic Reviews and Meta-Analyses) (Figure 1). In this review, we selected key articles based on hormonal contraception and female sexual function. PubMed and Cochrane were chosen as the main databases used due to the extensive contents of biomedical research they offer, their free access, and their ease of use. Our search term combinations were: “Hormonal contraception” AND “female sexual function” OR “female sexual dysfunction.” The filters “publication date: From 2000/01/01 to 2019/01/31” and “review” were applied in the search in order to limit the amount of material available. No language restrictions were applied. Similar and related articles that were considered of special interest for our review were also included, and they were compiled though cross-referencing. Similarly, some relevant clinical practice guidelines were included. The 64 papers that were included were chosen because they fit the topic of the review (presenting information about female sexual dysfunction, hormonal contraception, hormonal variations, and their relationship with female sexual function; directly treating the impact of hormonal contraceptives in female sexual function; or providing relevant information about the management strategies of female sexual dysfunction associated with the use of HCs). We reviewed six prospective observational studies, eight clinical trials, 19 cross-sectional studies, 22 reviews, and nine other works that include consensus and clinical practice guidelines. Most of the studies were carried out in European countries, although there were also studies carried out in the US, Asia, Australia, and South America. The population of the studies reviewed varied between 40 and 18,787, although in the case of clinical trials, the largest population analyzed was 600 subjects.

We summarized the findings and best practice recommendations for addressing a woman’s contraception and its potential association with sexual function. We excluded those articles that focused on male sexual dysfunction, menopause, and sexual dysfunction related to medical disease, such as oncological pathology. Every attempt was made to combine as much similar data as possible. Institutional review board approval was not needed for this review.

## 3. Results

### 3.1. Hormonal Contraceptives

The combined oral contraceptive (COC) was first approved in 1960. Since then, it has undergone many evolutions in dosage, hormone type, and regimen. It has been used by more than 100 million women worldwide and has the widest geographic distribution of any method of contraception [10]. In this section, we will provide detailed information about hormonal contraceptives in terms of the existing types, their hormonal composition, their mechanism of action, and the alterations in hormonal function that derive from them.

#### 3.1.1. Types

At present, there are twenty different contraceptive methods approved by the FDA [20], ten of which are female hormonal contraceptive methods: Eight are reversible contraceptive methods, and two are emergency contraceptive methods. In Table 1, we can see the different categories of hormonal contraceptives mentioned.

Reversible contraceptive methods include: Combined hormonal contraceptives (CHCs), progestin-only contraceptives, and intrauterine contraceptives (IUCs). COCs include the “pill” or combined oral contraceptives (COCs), the contraceptive patch, and the vaginal ring. When talking about progestin-only contraceptives, we can differentiate between progestin-only-pills (POPs), depot medroxyprogesterone acetate (DMPA), and the “implant” or single rod etonogestrel subdermal implant. IUCs include copper intrauterine devices (Cu-IUDs) and levonorgestrel-releasing intrauterine systems (LNG-IUS) [10,21,22]. Emergency hormonal contraceptives (ECs) are: Levonorgestrel of 1.5 mg (1 pill) or 0.75 mg (2 pills) and ulipristal acetate [20]. Permanent contraceptive methods that are approved by the FDA are: Sterilization surgery for women, a sterilization implant for women, and sterilization surgery for men [20].

#### 3.1.2. Hormones

The hormonal composition of hormonal contraceptives is based on progestins alone or on a combination of progestogens and estrogens [10,20,21,22,23,24]. Several different progestins are used in combined oral contraceptives (COCs). These progestins may also have estrogenic, antiestrogenic, androgenic, antiandrogenic, or antimineralocorticoid activity [10]. Most progestins are 19-nortestosterone derivatives. Progestins may be classified according to their chemical structure as an estrane (norethindrone, norethindrone acetate, ethynodiol diacetate) or as a gonane (LNG, desogestrel, norgestimate). In general, gonane progestins appear to be more potent than the estrane derivatives (smaller doses can be used), but other differences between the estrane and gonane compounds are difficult to characterize [10]. Table 2 shows the classification of progestogens used in hormonal contraception according to their androgenic potency. Among the contraceptive progestins available in the United States, norgestrel and levonorgestrel are the most androgenic; norethindrone and norethindrone acetate are less androgenic; and desogestrel, etonogestrel, norgestimate, dienogest, and drospirenone are the least androgenic [2]. Newer progestins (norgestimate and desogestrel) have little or no androgenic activity, whereas other progestins (cyproterone acetate, drospirenone, and dienogest) have antiandrogenic activity [10]. The varying progestational “potencies” attributed to different COC preparations are based on pharmacological experimental models. Many variables affect the potency of COCs (including dosage, bioavailability, protein binding, receptor binding affinity, and interindividual variability), making it difficult to extrapolate the results of isolated experiments to provide clinically relevant information in humans. There is no clear clinical or epidemiological evidence that compares the relative potencies of currently available COCs [10]. Systemic progestins may be associated with a loss of sexual desire due to the suppression of ovarian function and endogenous estrogen production [6]. Along the same line of reasoning, in their study about women’s self-reported sexual desire across natural cycles, Roney and Simmons observed that levels of salivary progesterone negatively predicted women’s sexual desire [25,26]. Furthermore, based on the findings by Grebe et al., effective dosages of progestin should be associated with a stronger positive linkage between women’s loyalty/faithfulness to their relationship partners and the frequency with which they engaged in sexual intercourse with their partners [26,27]. However, contraceptive pills with progestogens with antiandrogenic effect do not affect sexual desire, according to some reports [28,29]. In recent studies, drospirenone and dienogest have reported a positive effect on sexual response as well as attraction, desire, satisfaction, and coital frequency [28,30], perhaps due to the ability to reduce the activity of 5-alpha reductase [31].

With regard to estrogens as hormonal components of hormonal contraceptive methods, three types of estrogens are used in COCs (as it can be seen in Figure 2): Ethinylestradiol (EE), estradiol valerate (E2V), and 17 beta-estradiol (E2). E2V is rapidly metabolized to E2 [10]. Due to its biochemical structure, estradiol has less impact on the synthesis of hepatic proteins than ethinyl estradiol, which is likely to result in a better metabolic and vascular profile [3]. The new formulations of launched COCs have lower doses of estrogen, and EE has been replaced by more “physiological” forms of estrogen, such as 17β-estradiol (E2) or E2-Valerate (E2 V) [32]. There is some evidence to suggest that estrogens play an essential role in female sexuality, and prior research has found that declining sexual functioning in women is most closely related to declining estrogen levels [6,33] Similarly, levels of salivary estradiol positively predicted women’s sexual desire, conversely to progesterone [25,26]. Regarding loyalty and faithfulness, dosages of estradiol should predict a weaker positive linkage between women’s loyalty/faithfulness to their relationship partners and frequency of sexual intercourse (not including masturbation and sexual fantasies; independently of androgenicity of sexual hormones) [26,27].

#### 3.1.3. Mechanism of Action of Hormonal Contraceptives

In Table 1, we can see a summary of the different categories of hormonal contraceptives mentioned with their respective mechanism of action of hormonal contraceptives. The mechanism of action of hormonal contraceptives depends on their hormonal composition and the route of administration.

Combined hormonal contraceptives (CHCs) encompass oral contraceptives (pill), patch, and the vaginal ring. Their mechanism of action is similar.

With regard to combined oral contraceptives (COCs), they have multiple mechanisms of action due to both their estrogenic and progestational components: The suppression of pituitary gonadotropin secretion (inhibiting ovulation), the increase of cervical mucus viscosity (impairing sperm transport), the suppression of the luteinizing hormone (LH), and the impairment of ovulation [10].

The patch is a 20 cm^2^ square matrix system that delivers 200 mg of norelgestromin (the primary active metabolite of norgestimate) and 35 mg of ethinylestradiol (EE) daily to the systemic circulation. Following the first application of the patch, serum hormone levels increase gradually over the first 48–72 h, reach a plateau, and then remain constant during the remainder of the 21-day period. Compared with COC, plasma hormone levels remain constant, and the peak levels are lower because first-pass hepatic metabolism and gastrointestinal enzyme degradation are avoided. Curiously, although peak levels are lower, the area under the curve, which represents overall EE exposure, is larger. One patch is applied weekly for three consecutive weeks, followed by a one patch-free week. The patch can be placed on one of four sites: The buttocks, upper outer arm, lower abdomen, or upper torso, excluding the breast [10].

The ring releases 15 mg of EE and 120 mg of the progestin etonogestrel (ENG) (the active metabolite of desogestrel) per day, which is absorbed through the vaginal epithelium. Serum hormone levels increase immediately after ring insertion and then decrease slowly over the cycle [10]. The vaginal route is an ideal method of drug administration, and the advantages of this method are well established. By avoiding gastrointestinal absorption and the hepatic first-pass effect, the vaginal administration of contraceptives enables the use of lower hormonal doses and the achievement of steady drug concentrations [34].

There is another group of hormonal contraceptives only composed of progesterone. This group can include the progestin-only pill, depot medroxyprogesterone acetate (DMPA), and the etonogestrel implant. Progestin-only pills (POPs, the “mini-pill”) provide reliable, reversible contraception and have very few contraindications. The main mechanism of action is the alteration of the cervical mucus (more viscid, less copious) and the inhibition of sperm penetration. Negative luteinizing hormone (LH) feedback leads to the suppression of ovulation in up to 50% of users. POPs containing desogestrel may inhibit ovulation more consistently [21].

DMPA is administered intramuscularly at three-month intervals (every 12–13 weeks) and is thus considered a long-acting reversible contraceptive (LARC) by some and a short-acting reversible contraceptive (SARC) by others. DMPA works primarily by inhibiting the secretion of pituitary gonadotropins, thereby suppressing ovulation. Women enter a hypoestrogenic state, and their progesterone is low due to anovulation. DMPA also increases the viscosity of cervical mucus (minor mechanism of action) and induces endometrial atrophy [21].

The single-rod etonogestrel subdermal implant (Implanon/Implanon NXT/Nexplanon) is a LARC. The single-rod implant contains 68 mg of the progestin etonogestrel (ENG) and provides contraception for three years. The ENG implant works primarily by inhibiting ovulation and consistently does so until the beginning of the third year of use. Ovarian activity, including estradiol synthesis, is still present. The ENG implant causes a thickening of the cervical mucus and changes in the endometrial lining [21].

The last group is formed by intrauterine contraceptives (IUCs). This group includes copper intrauterine devices (Cu-IUDs) and levonorgestrel-releasing intrauterine systems (LNG-IUS). Only LNG-IUS are explained in this section, because Cu-IUDs do not have a hormonal component. The chief mechanism of action of all IUCs is the prevention of fertilization; they may also have post-fertilization effects, including the potential inhibition of implantation. The LNG-IUS produce a weak foreign body reaction and endometrial changes that include endometrial decidualization and glandular atrophy. The primary mechanism of action is via changes in the amount and the viscosity of cervical mucus, which acts as a barrier to sperm penetration. Ovulation is likely inhibited in some women, but it is preserved in most study subjects. Endometrial estrogen and progesterone receptors are suppressed, which results in changes in bleeding patterns and may contribute to its contraceptive effect [22].

#### 3.1.4. Hormonal Alterations of Hormonal Contraceptives and Their Influence on Female Sexual Function

In contrast to animal species in which linear relationships exist between hormonal status and sexual behavior, sexuality in the human population is remarkably complex and is not determined so simply by the level of sexual steroids [29].

Hormonal contraceptives (HCs) are responsible for a decrease of circulating androgen levels [1,2,29,35], as well as a decrease of the baseline serum levels of estradiol [6,29,35] and progesterone [35] and the inhibition of oxytocin functioning [35]. However, the concentrations of the follicle-stimulating and luteinizing hormones are similar in freely cycling women and in women using HCs [35]. Decreased circulating androgen levels with oral combined hormonal contraceptive (CHC) use, and its negative effects on sexual life, occur by two mechanisms, as follows: (1) An oral CHC increases sex hormone-binding globulin (SHBG) and decreases free testosterone, and (2) androgen production from the ovary is suppressed with an oral CHC. This antiandrogenic effect may be magnified with an oral CHC containing an antiandrogenic progestin [2]. Thus, all CHCs are antiandrogenic, although some formulations, depending on the specific progestin, are more so than others. The patch and the vaginal ring are more antiandrogenic than the pill [1]. As expected, the baseline serum levels of estradiol and progesterone are significantly higher in freely cycling women than in women using an HC. Nevertheless, the concentrations of the follicle-stimulating and luteinizing hormones are similar in both groups [35]. In respect of oxytocin, its functioning is likely to be altered by this variation in the peripheral estradiol and progesterone levels that were found to be altered in women using HCs, and, therefore, a potential mechanism could be related to the direct binding of progesterone to oxytocin receptors (OXTRs), thereby inhibiting OXTR functioning.

The association between hormones and sexuality is multidimensional, as several hormones are important in the regulation of sexual behavior [29].

Though some evidence shows that testosterone has a role in sexual function for women, these conclusions are derived primarily from studies involving postmenopausal women reporting sexual dysfunction [2]. It has been established that sexual desire, autoeroticism, and sexual fantasies in women depend on androgen levels [29]. However, the relevance of changes in androgen levels for an individual woman is unclear, and some women may be more sensitive to androgen level alteration than others [2]. The review by Casey et al. mentioned that most of the studies showed alterations in SHBG and testosterone levels; however, an overall lack of association was found between CHCs and sexual desire [2]. In other studies, decreased levels of estrogen and testosterone in older women have been associated with decreased libido, sensitivity, and erotic stimuli [29]. In addition, it has been found that patients using birth control pills may present with decreased libido. On the other hand, there are reports that suggest that progestogens with antiandrogenic effects in contraceptive pills do not affect sexual desire [29]. While there is conflicting evidence concerning a link between progestins and libido, there is some evidence to suggest that estrogens play an essential role in female sexuality. In this respect, prior research has found that declining sexual functioning in women is most closely related to declining estrogen levels [6].

Finally, with regard to oxytocin, Scheele et al. [35] describe in their work the possible functional implications of oxytocin in female sexuality and the alterations that occur in women who take hormonal contraceptives. Multiple lines of evidence suggest that the hypothalamic peptide oxytocin (OXT) is a key factor modulating pair-bonding behaviors, which means a strong affinity that develops in humans and some species between a mating couple.

In humans, peripheral OXT concentrations are significantly higher in new lovers compared with singles. Likewise, OXT reduces jealousy ratings and neural responses in an imagery task of sexual partner infidelity. OXT also increases the arousal induced by infant photos in nulliparous women and promotes responsiveness to infant crying and laughter by reducing activation in anxiety-related neural circuits. Moreover, OXT has been found to increase the intensity of orgasm and contentment after copulation. Nevertheless, OXT seems to not have an effect on vital signs. The results of the research by Scheele et al. [35] indicate that endogenous OXT concentrations at baseline positively predicted striatal responses to the romantic partners’ faces in all female participants. This mechanism was disturbed in those women using an HC, indicating that the partner-specific modulatory effects of OXT are antagonized by gonadal steroids. HC use alters women’s pair-bonding behavior (evident in decreased attractiveness ratings of masculine faces), reduced neural response to the expectation of erotic stimuli (a preference shift towards olfactory cues of genetic similarity), and increased sexual jealousy. Furthermore, women who use an HC while choosing partners are more likely to initiate an eventual separation, and wives who discontinue HC use tend to be less satisfied with marriage if they perceive their husband’s face to be less attractive. On the other hand, women prefer masculine faces and exhibit higher levels of intersexual competition related to attractiveness at peak fertility in the menstrual cycle; however, these cyclical shifts were found to be diminished in women using an HC. In conclusion, OXT interacts with the brain reward system to reinforce partner value representations in both sexes, a mechanism which may significantly contribute to stable pair-bonding in humans and appears to be altered in women using an HC.

### 3.2. Sexual Dysfunction

To talk about the effects on sexual function, it is first convenient to define the concept of sexual dysfunction, as well as the types of female sexual dysfunction that are currently described. In this section, the methods used and validated to quantify the degree of sexual dysfunction are also briefly discussed. In addition, an estimate of the prevalence of sexual dysfunction in the female population of childbearing age is shown.

According to the DSM-5 (Diagnostic and Statistical Manual of Mental Disorders, Fifth Edition), sexual dysfunctions are a heterogeneous group of disorders that are typically characterized by a clinically significant disturbance in a person’s ability to respond sexually or to experience sexual pleasure [18,36]. On the contrary, we would define “sexual health” as a state of physical, emotional, mental, and social well-being related to sexuality; it is not merely the absence of disease, dysfunction, or infirmary. Sexual health requires a positive and respectful approach to sexuality and sexual relationships, as well as the possibility of having pleasurable and safe sexual experiences, free of coercion, discrimination, and violence [37].

Therefore, optimal sexual function transcends the simple absence of dysfunction [18]. In this regard too, multiple studies have shown a strong positive association between sexual function and the health-related quality of life [18]. Having said that, it can be gathered that the female sexual function is complex and multifactorial, and it is influenced by many biological, psychological, and environmental factors [2,5,18,29]. Therefore, a complete understanding of women’s sexual function requires the individual assessment of these factors. The biopsychosocial approach recognizes that biological, psychological, interpersonal, and sociocultural factors can all affect female sexual function, and these factors interact with each other in a dynamic system over time. Biological factors may include hormonal changes that affect the libido or medical/anatomical problems that affect genital sexual response. Psychological factors include mood symptoms, like depression or anxiety, or negative behaviors such as critical self-monitoring during sexual activity. Some examples of interpersonal factors include general satisfaction in the woman’s relationship with her partner, which is closely tied to overall sexual satisfaction, as well as quality of communication in the relationship. Finally, some sociocultural factors to consider include the woman’s attitudes about menopause and aging, as well as religious, cultural, and other social values regarding sex [18].

When assessing alterations of sexual function possibly related to hormonal contraceptives, other factors that may also affect it should be taken into account. For example, sex hormones (mainly low levels of estradiol), physical and mental well-being, availability of a partner, feeling for her partner, illness and its treatments, changes in social circumstances, and low socioeconomic status could have an impact on women’s desire and sexual responsiveness [5,18]. Therefore, there are several factors that can affect female sexual function which should be explored by health providers for an adequate diagnostic and therapeutic approach to sexual dysfunction. However, there are studies that show in their results that sexual health is not a widely explored area for health providers in general. Mercer et al. showed that only 21% of women with persistent sexual problems discuss it with their healthcare provider [18,38]. Furthermore, a recent survey in the USA reported that the majority of gynecologists routinely ask patients about their sexual activities, but most other areas of patients’ sexuality, such as sexual problems, including pleasure and satisfaction, are not routinely discussed [34,39].

Theoretical models of women’s sexual response can provide a framework for a better understanding female sexual dysfunction. Three of these models are briefly explained here. First, according to the Masters–Johnson model, sexual response progresses predictably and linearly from excitement to plateau, orgasm, and resolution. The main focus of this model is on the physical response of the genitals. Secondly, Helen Singer Kaplan noted that many individuals had problems with sexual desire, denoting the importance of desire to sexual response. In the 1970s, she modified the Masters–Johnson model to a three-phase model of desire, excitement, and orgasm. Thirdly, in 2000, Rosemary Basson and colleagues proposed an alternative circular model of female sexual response. This model has several distinguishing features. On the one hand, spontaneous desire (or “sexual drive”) on the part of the woman is not always the starting point for sexual activity. On the other hand, this model emphasizes that sexual stimuli often precede physical arousal and desire, and sexual arousal and desire often co-occur. Finally, the Basson model acknowledges that both physical and emotional satisfaction are important outcomes of engaging in sexual activity. This physical and emotional satisfaction can lead to higher emotional intimacy, which, in turn, can lead to greater receptivity and seeking out of sexual stimuli—hence, the circular model [18].

There has been debate regarding which model best reflects the experiences of women. In a study of 133 women, most of whom were in their 40s and 50s, women who had Female Sexual Function Index (FSFI) scores falling into the “dysfunctional” range and postmenopausal women were more likely to endorse the Basson model [18,40].

With the concept of sexual dysfunction now developed, we may now discuss the types of sexual dysfunction that are described. Four types of female sexual dysfunction are currently recognized: (1) Female orgasmic disorder, (2) female sexual interest/arousal disorder, (3) genito-pelvic pain/penetration disorder, and (4) substance/medication-induced sexual dysfunction. In order to quantify sexual dysfunction in a fairly objective way, there are two commonly used instruments in sexual function studies: The Female Sexual Function Index (FSFI) and the Female Sexual Distress Scale-Revised (FSDS-R) [18]. The Female Sexual Function Index (FSFI) is a 19 item scale with six domains: Desire, arousal, lubrication, orgasm, pain, and satisfaction. In this scale, questions are graded on a Likert scale, and domains are weighted and summed to give a total score ranging from 2–36, with a cutoff of less than 26.55 suggesting sexual dysfunction. The FSFI has been validated in multiple languages, across age groups, and for multiple sexual disorders [18,41].

Why is it important to read up on sexual dysfunction? Sexual problems are common, estimated to affect 22–43% of women worldwide [18]. Overall, 27% of all reproductive-age US women (aged 18–44 years) report sexual dysfunction, with low sexual desire being the most common, and 10.8% of these women also experience related distress [2]. The prevalence of sexual dysfunction peaks at midlife, with 14% of women aged 45–64 reporting at least one sexual problem associated with significant distress [18]. The proportion with a notable or severe problem in desire, arousal, activity, or satisfaction ranges from 19–25% [5].

### 3.3. The Effects of Hormonal Contraceptives on Sexuality

This section presents different results found in the literature about the effects of hormonal contraceptives (HCs) on female sexuality (including results that advocate for positive or negative effects or the absence of sexual effects). It also discusses the peculiarities of the different types of HCs on sexuality.

#### 3.3.1. Hormonal Contraceptives Do Not Have Sexual Effects

Some studies have found no change in sexual function with some hormonal contraceptives (HC) [2,3,6,10,42,43,44,45,46]. A recent systematic review of 36 studies involving more than 13,000 women reported no significant changes in sexual desire with the use of oral combined hormonal contraception (CHC) [43]. Another study [47] also reported high satisfaction rates with both LNG-IUS and copper IUC but no difference in sexual function overall or within psychological domains. In another recent study, no association was found between any LARC method and sexual satisfaction scores [48].

On the other hand, Reed et al. explored the relationship between oral contraceptive (OC) use and the risk of developing vulvodynia [49]. Further analysis showed no association between vulvodynia and previous OC use (HR 1.08, 95% CI 0.81–1.43, *p* = 0.60). In a study by Iliadou et al. [50], patients reporting mixed urinary incontinence (MUI) were divided into three groups according to contraceptive use. Of 196 women with MUI, 16 were currently using OC, and 178 reported no current use. Among the 8493 controls, 6321 were not using OC, and 2056 were (*p* < 0.0001). A systematic review of the literature found that sex drive is unaffected in most women taking OC, 3.5% of women taking OC reported a decrease in sexual desire, 12.0% reported an increase, and most of them (84.6%) reported no change [43]. However, the effects of other forms of hormonal contraception on sex drive have not been studied as comprehensively as OC [1].

#### 3.3.2. Hormonal Contraceptives Have Sexual Effects

##### Positive Effects

According to the studies reviewed, hormonal contraceptives have a series of non-contraceptive effects which can influence and improve different areas of female sexual function. Some of these non-contraceptive effects are: Relief of gynecologic pain [1]; improved appearance, self-confidence, and self-esteem [2]; decrease of anxiety and discomfort [2]; loss of fear of having an unwanted pregnancy [6]; more stable levels of hormones throughout the cycle [51]; and less bleeding with the consequent lower risk of anemia [51]. All these effects contribute to the well-being of women and, consequently, to a possible improvement in the female sexual function. Similarly, hormonal contraceptives have described positive effects on some areas of female sexuality. The most frequently affected areas are: Sexual desire, orgasm number and intensity, satisfaction, and arousal. As mentioned, HCs may help to eliminate the fear of pregnancy, presumably providing a more relaxed and enjoyable sexual experience [1]. Similarly, it is reasonable to consider that an improved appearance would promote self-confidence and increase self-esteem, thereby having a positive effect on sexual function [2]. In a comparison between the vaginal ring, an oral CHC containing a third-generation progestin, subdermal contraception, and no hormonal contraception (control group), the three groups using an HC had increased positive indicators of sexual function (sexual interest and fantasies, orgasm number and intensity, and satisfaction) and decreased negative indicators (anxiety and discomfort). The same results were obtained in a comparison between etonogestrel implant and no contraception [2,52]. LNG-IUS have also been positively associated with sexual desire, arousal, orgasm, and overall sexual function compared with no contraception [2,53].

Furthermore, it may be advantageous for women to have more stable levels of hormones throughout the cycle. Because of the monthly fluctuations in estrogens, progesterone and androgens are associated with a range of symptoms, both genital (i.e., vaginal bleeding, heavy menstrual bleeding (HMB), dysmenorrhea, and pelvic pain) and systemic (i.e., depression, fatigue, headache, irritable bowel symptoms (IBS), asthma, and allergy), triggered by a local and systemic rise in inflammatory molecules released by mast cells when estrogen levels drop [51].

##### Negative Effects

To begin with, diminished sexual pleasure experienced by some women who use hormonal contraceptive methods may also be a barrier for their use [54], and this could imply an increase in the woman’s vulnerability to unintended pregnancy [54]. Consequently, it is important to keep in mind that hormonal contraceptives could have associated side effects that have an influence on female sexual function. Some of these effects could be: Vaginal dryness [2,10,51], a decrease of lubrication [2,51], and pelvic floor symptoms such as dyspareunia [3,51], urinary incontinence, vestibulodynia, and interstitial cystitis [3]. COCs have been also associated with long- and short-term anatomical changes, such as atrophic vulvovaginitis and a decrease of thickness of the labia minora and vaginal introitus area [1]. Negative effects on some areas of female sexuality have been described with HCs, such as: Decreased sexual desire [2,6,10,54], frequency of intercourse [2,54], arousal [2,54], pleasure [2,54], orgasm [2,54], sexual thoughts [54], interest, and enjoyment [6,54].

In contrast to the above section, Elaut et al. [46] and Li et al. [55] defend in their studies that desire and coital frequency naturally increase around ovulation and premenstrually, and COC-associated ovulation inhibition and cycle regulation may blunt this effect, with the corresponding negative impact on libido [10]. Furthermore, longer durations of oral CHC use and younger ages at initiation have been associated with a higher relative risk of vestibulodynia [2], with the resulting negative impact on female sexual function.

#### 3.3.3. Effects on Sexual Function According to the Type of Hormonal Contraceptive

Combined oral contraceptives are widely studied. Nevertheless, other hormonal contraception methods have fewer studies about their influence on sexual function. In this section, the results obtained from the studies reviewed for each type of hormonal contraceptive will be presented. Table 1 shows a summary of this information.

##### Contraceptive Patch

Concerning patch-related sexual effects, this could be considered the most innocuous CHC. Gracia et al. [56] found that among recent COC users, slight increases in sexual function scores were noted with patch use. However, they concluded that for both products, these changes are not likely to be clinically significant [1,34]. Therefore, it would be advisable to expand the research in this regard.

##### Contraceptive Ring

With regard to ring-related sexual effects, there are mixed results. On the one hand, two studies showed a decrease in sexual function with vaginal ring compared with COCs [56,57], and one study showed similar results but compared with the patch [58]. However, an improvement in sexual function including sexual desire, fantasies, and satisfaction, accompanied by a reduction of sexual distress, has been described with the vaginal ring [1,2,10,34]. In another study [34], compared with nonusers of hormonal contraception, both vaginal ring and COC users reported significant improvements for anxiousness, sexual pleasure, frequency and intensity of orgasm, satisfaction (all *p* < 0.001), sexual interest, and complicity (*p* < 0.01). However, only women in the vaginal ring group reported a significant increase in sexual fantasies (*p* < 0.001 versus nonusers), while ratings for sexual interest and complicity were significantly higher in ring users versus COC users [34]. As suggested by the researchers, these data indicate that both oral and vaginal contraception seem to improve to some extent the sexual life of women and their partners, whereas the vaginal ring seems to exert a further beneficial effect on the psychological aspects of sexual functioning [59].

Vaginal contraception offers many benefits, including high efficacy, good tolerability, ease of use, once-a-month dosing, and a favorable pharmacokinetic profile, with the added benefits of positive effects on the vaginal microbiome and on sexual parameters [34]. In addition, good cycle control and less fluctuating serum hormonal levels could contribute to the high degree of users’ acceptability and satisfaction. Most importantly, a discussion about the vaginal delivery of contraceptive hormones offers the opportunity to stimulate an open dialogue about vaginal functions, thus ultimately contributing to enhancing women’s sexual well-being and reproductive health [34]. Consequently, it could be a good hormonal contraceptive option.

##### Depot Medroxyprogesterone Acetate (DMPA)

DMPA is a highly effective method of contraception. It has been used as a contraceptive agent since 1967 by millions of women worldwide, particularly in less developed regions [21]. In respect of DMPA-related sexual effects, there are mixed results. Despite decreased libido being a common complaint among DMPA users and the fact that progestins have been observed to decrease interest in sex [6], positive sexual effects are also described with this method [6,60]—some reviews even reveal that DMPA is unlikely to be associated with sexual function in women [1,2,6]. However, further research would be needed to support these claims.

##### Etonogestrel Implant

Etonogestrel implant-related sexual effects are described as negative effects. It has been associated with a lack of interest in sex, a decreased libido, and a reduced sex drive. In addition, a decreased libido has been observed as a significant cause for implant discontinuation [1,6].

##### Levonorgestrel-Releasing Intrauterine Systems (LNG-IUS)

Intrauterine contraceptives (IUCs) are long-acting reversible contraceptive (LARC) methods that are used by over 150 million women worldwide. IUCs are highly effective methods of contraception that can be used by women of all ages. Rates of IUC use vary throughout the world, from a maximum of 41% in China to a minimum of 0.8% in sub-Saharan Africa [22]. They have generally been associated with positive sexual effects. They have been reported to improve desire, sexual function, and arousal [1,2,60]. Moreover, they seem to improve the health-related quality of life through the improvement of dysmenorrhea and symptoms in patients with endometriosis and adenomyosis, among other things [22].

#### 3.3.4. Other Non-Hormonal Methods of Contraception and Their Effect on Sexual Function

##### Copper Intrauterine Devices (Cu-IUDs)

There has been no evidence to suggest that the copper IUD is associated with an altered libido [6].

##### Vasectomy/Tubal Ligation

As a non-hormonal contraceptive method, the effect of sterilization on sexual function extends beyond a simple hormonal effect into the psychological aspects of permanent pregnancy prevention, whether positive (i.e., relief and comfort in the knowledge that sexual activity will not result in pregnancy) or negative (i.e., regret that pregnancy is no longer possible) [2].

##### Nonuse of Contraception

Female sexual function is complex and multifactorial and is influenced by many biological, psychological, and environmental factors [2,5,18,29]. Therefore, a complete understanding of women’s sexual function requires the individual assessment of these factors. Consequently, sexual dysfunction does not have to be associated with hormonal contraception. The use of no contraception was associated with a higher rate of the FSD than the use of either CHCs or nonhormonal methods. Furthermore, lower rates of sexual dysfunction were noted among women using either copper IUC (21%) or a levonorgestrel intrauterine systems (LNG-IUS) (10%) than among women using no contraception (35%). Among other reasons, diminished sexual function perceived to be related to contraception may lead to the nonuse of effective contraception, and, conversely, the nonuse of contraception may in itself be a factor in sexual dysfunction, perhaps owing to concerns about unintended pregnancy [2].

#### 3.3.5. The Sexual Side Effects of Hormonal Contraceptives are not Well Studied

Existing evidence for an association between sexual dysfunction and contraception is inconsistent, and additional research is needed [2]. Findings from studies comparing women using non hormonal contraception with those using hormonal methods have shown mixed results [2]. The sexual side effects of hormonal contraceptives are not well studied, particularly with regard to their impact on libido [1]. Similarly, there is no clear information about the effect of HCs on pelvic symptoms and sexual function, nor on how they affect a woman’s quality of life in relation to bowel and bladder symptoms, regardless of period control and menstrual bleeding. Moreover, the association between COC use and the presence of any type of urinary incontinence (UI) is unclear, and results suggest that the effect of current COC use on dyspareunia per se is inconsistent [3].

Healthcare care providers must be aware that hormonal contraceptives can have negative effects on female sexuality so they can counsel and care for their patients appropriately [1]. In order to better evaluate any possible effect on mood or libido, practitioners should assess patients prior to initiation of hormonal contraception to establish their baseline [60]. The lack of consistency in findings highlights the complex and multifactorial nature of female sexual function and focuses on the need for a comprehensive approach to management [2].

### 3.4. Management Strategies for Sexual Dysfunction Secondary to Hormonal Contraceptives

This section approaches the therapeutic possibilities for female sexual dysfunction described in the literature. In addition, some keys are given for the management of sexual dysfunction secondary to hormonal contraceptives (Figure 3).

First, when addressing a new sexual complaint, a thorough history using a biopsychosocial approach should be undertaken (Table 3) [18], including an assessment of any current or past psychiatric disorders; medication use and health problems; a history of emotional, physical, or sexual abuse; beliefs and attitudes regarding sex, menopause, and aging; and body image concerns. Particular attention should be paid to symptoms of depression, anxiety, and sleep problems, all of which are common during the menopause transition. Providers should inquire about alcohol or drug use, as substance use disorders are also associated with sexual dysfunction. Any health or sexual problems affecting the woman’s sexual partner(s) should also be explored. Providers should inquire about relationship discord or communication issues, and if present, recommend therapy with a certified and specialized therapist [18]. A multidisciplinary approach to the management of female sexual dysfunction (FSD) is suggested, particularly when multiple contributing or complicating factors are identified, and this may consist of consultations with other professionals, such as a sex therapist, a pelvic floor physical therapist, and a sexual health specialist [2].

Second, lifestyle counselling should be given by the health providers. General lifestyle counselling that may be useful for all types of female sexual dysfunction include recommending setting aside time for connecting with one’s partner, increasing the woman’s exposure to sexual stimuli such as erotic literature or films, encouraging the maintenance of a healthy weight, ensuring adequate physical activity and sleep, enhancing skills for coping with stress, and recommending books women can use for self-education (Table 4) [18].

When choosing a new hormonal contraception method, health care providers (HCPs) should give information about all available methods in order to make a shared decision [34]. In the Contraceptive CHOICE Project, a prospective cohort study of 10,000 women 14–45 years who want to avoid pregnancy for at least one year and are initiating a new form of reversible contraception, 47% of women who had an interest in a CHC method selected a different method than the one they originally intended to use after receiving counselling about several CHC methods, including the pill, patch, and ring. Awareness of the decision-making factors that affect women’s choices regarding methods of contraception may enable HCPs to make more informed recommendations that are targeted to the needs of each of their female patients [4]. The prescription of a contraceptive method is a great opportunity to clarify the multidimensional components of sexual health, including elements of anatomy and physiology of the sexual response [34].

Few clinical remedies or recommendations exist for women experiencing HC-related sexual side effects [54]. Unfortunately, no guidelines exist for the management of sexual dysfunction potentially associated with CHCs in reproductive-age women [2]. As such, when CHC-related female sexual dysfunction is suspected, the recommended therapy is discontinuation of a combined hormonal contraceptive, with consideration of an alternative method of contraception, such as LNG-IUS, a copper IUC, a etonogestrel implant, the permanent sterilization of either partner when future fertility is not desired, or a contraceptive ring (for women who prefer a CHC for cycle control and no contraceptive benefits) [2]. The ring appears to be a reasonable alternative to an oral CHC for women with sexual function concerns. Likewise, LARC methods also appear to be a reasonable alternative [2]. Nevertheless, switching to another combined oral contraceptive may provide some benefit, but there is no clear difference between androgenic or non-androgenic progestins [10]. In addition, the combination of dehydroepiandrosterone (DHEA) and an OC was not associated with improvements in sexual function, and it further negated the benefit of OCs on acne [2]. When COC-related female sexual dysfunction is suspected, another possible option could be to consider formulations with a shorter hormonal free interval (HFI). Formulations with a shorter HFI (24/4 and 26/2) have recently been developed with the aim of offering a reduction in hormone withdrawal-associated symptoms together with a more powerful ovarian suppression. Estradiol valerate/dienogest (E2V/DNG) is administered on a 26/2 regimen and has been shown to offer a high contraceptive efficacy, an improvement in hormone withdrawal-associated symptoms (including but not limited to headache and pelvic pain), and an improvement in sexual function [51,61]. In conclusion, the best contraceptive is one that fulfills women’s needs with acceptable side effects and at an affordable price in different settings [32].

Other options to improve HC-related sexual dysfunction could be vaginal lubricants and moisturizers. They are the first-line treatment for vaginal dryness and consequent dyspareunia [2], side effects that are frequently associated with hormonal contraceptives, mainly with combined oral contraceptives. The majority of women participating in a daily study reported positive perceptions of lubricant use, including increased pleasure and comfort [62]. Sharing information on the high frequency of use and positive results experienced across age-groups may be helpful in counseling reproductive-age women about using lubricants [62].

Furthermore, concerning other possible strategies against sexual dysfunction, some studies show positive results on female sexual function with exogenous testosterone [2,18,29], exogenous estrogens [2,6], dehydroepiandrosterone (DHEA) [10,29], tibolone [29], bupropion, and sildenafil [18]. It appears that supraphysiological serum testosterone levels may be necessary to yield any benefit on sexual desire and arousal [18]. The use of compounded testosterone products for transdermal use is on the rise, but these products are not FDA-approved [18], and they can be associated with several side effects. Meanwhile, testosterone therapy in postmenopausal women has been associated with improvements in multiple dimensions of sexual function, including sexual desire, subjective arousal, vaginal blood flow, and frequency of orgasm [2]. Testosterone released from patches has also been described to produce positive effects on mood and sexual behavior and to increase bone mass significantly [63].

With regard to hormonal therapy with exogenous estrogens, results are controversial. On the one hand, exogenous estrogens have been shown to be an effective treatment for low libido and hypoactive sexual desire disorder [6], and, on the other hand, hormone therapy (estrogen with or without progesterone) does not appear to have a significant impact on sexual function, with the exception of vaginal estrogen in women with the genitourinary syndrome of menopause [18]; that is to say, hormonal therapy with estrogen is efficient with regard to genital atrophy, but it is not efficient in regard to sexual desire [29].

Furthermore, although dehydroepiandrosterone (DHEA) supplementation could have positive effects on the female libido [29] by restoring androgen levels in COC users, there is minimal evidence that this correlates with improved sexual functioning [10]. There is also evidence that bupropion and, to a lesser extent, sildenafil, are effective for treating antidepressant-induced sexual dysfunction in women, although some conflicting evidence exists [18].

To conclude, even today, most of the contraceptives available on the market and those currently undergoing research and development interfere with ovulation or follicular development and also affect women’s steroid production [32]. This mechanism of action is associated with several side effects, negative sexual effects included, that could be avoided by new contraceptives strategies. For that purpose, research conducted over the past few decades has provided more information on gamete physiology and interaction, offering new opportunities for the development of novel contraceptives that could act by interfering with the process of gamete interaction or with the chemo-attraction or chemo-repulsion of spermatozoa to the fertilization site without affecting the hormonal system [32].

## 4. Discussion

As discussed in the review above, hormonal contraception (HC) has made a difference in the control of female fertility since its approval by the FDA almost 60 years ago, and it is also widely used in the female population of child bearing age. Side effects, such as sexual dysfunction, may be sufficient reasons for the discontinuation of this contraceptive method. This represents an increase of the risk of unwanted pregnancy, with the possible worsening of women’s wellbeing. However, female sexual function is complex and multifactorial and, despite an association between hormonal contraception and sexual dysfunction having been described in the past, the evidence on that topic is inconsistent.

Sexual problems are common, estimated to affect 22–43% of women worldwide [18], and influencing some types of female sexual dysfunction such as orgasm, sexual interest/arousal, and genito-pelvic pain. As a consequence of the multiple medications on sexual functioning, a specific category has been included in the new American DSM-5 classification system: Substance/medication-induced sexual dysfunction) [18]. As said above, female sexual function is complex and multifactorial, and a biopsychosocial approach to sexual problems is recommended. It could be said that an HC can influence female sexual function in two different ways. On the one hand, an HC could have a negative influence on sexual function as a biologic factor, because HC use has been associated to hormonal changes. On the other hand, an HC could have a positive influence on sexual function, in psychological terms, since HC use has been associated with an improvement in mood symptoms and self-perception. Different options for hormonal contraception exist. There are three main groups: Combined hormonal contraception (pill, patch and vaginal ring); progestin-only contraceptives (POPs, DMPA, and implant); and intrauterine devices (LNG-IUDs). The hormonal composition of hormonal contraceptives is based on progestins alone or on a combination of progestogens and estrogens. Apparently, norgestimate and desogestrel, among progestogens, and 17B-estradiol (E2) and E2-valerate (E2V), among estrogens, have a profile less associated with side effects than the others in their respective groups.

The association between hormones and sexuality is multidimensional, as several hormones are important in the regulation of sexual behavior [29]. Hormonal contraceptives (HCs) seem to be responsible for a decrease of circulating androgen levels [1,2,29,35], baseline serum levels of estradiol [6,29,35], and baseline serum levels of progesterone [35], as well as the inhibition of oxytocin functioning [35]. However, the concentrations of the FSH and LH were similar in freely cycling women and in women using an HC [35]. These hormonal alterations can be translated into negative effects on the female sexual function, with reports of a decrease of the libido, increased sexual jealousy, and alterations on women pair-bonding behavior. It has been established that sexual desire, autoeroticism, and sexual fantasies of women depend on androgen levels [29]. However, the relevance of changes in androgen levels for an individual woman is unclear, and some women may be more sensitive to androgen level alteration than others [2]. Furthermore, while there is conflicting information concerning a link between progestins and libido, there is some evidence to suggest that estrogens play an essential role in female sexuality [6]. On the other hand, multiple lines of evidence suggest that the hypothalamic peptide oxytocin (OXT) is a key factor modulating pair-bonding behaviors, and it has been found to increase the intensity of orgasm and satisfaction after copulation. This mechanism was disturbed in those women using an HC, indicating that the partner-specific modulatory effects of OXT are antagonized by gonadal steroids. So, it could be said that HC use alters women’s pair-bonding behavior, reduces neural response to the expectation of erotic stimuli, and increases sexual jealousy.

Despite an association between hormonal contraception and sexual function having been described, there are contradictory results between different studies in this respect. Some studies have found no change in sexual function with hormonal contraceptives (HCs) [2,3,6,10,42,43,44,45,46].

According to the studies reviewed, hormonal contraceptives have a series of non-contraceptive effects, which can be related to an improvement on different areas of female sexual function such as sexual desire, orgasm number and intensity, satisfaction, and arousal. All these effects contribute to the well-being of women and, consequently, to a possible improvement in the female sexual function.

By contrast, HCs could be associated with side effects that have an influence on female sexual function. Negative effects on some areas of female sexuality have been described with hormonal contraceptives, such as sexual desire [2,6,10,54], frequency of intercourse [2,54], arousal [2,54], pleasure [2,54], orgasm [2,54], sexual thoughts [54], interest, and enjoyment [6,54].

Combined oral contraceptives are widely studied, and most studies are based on COCs or used them as a comparative method of contraception. Nevertheless, other hormonal contraception methods have fewer studies about their influence on sexual function. There are mixed results with ring- and DMPA-related sexual side effects. The patch could be considered the most innocuous CHC regarding sexual side effects. The implant has been associated with negative sexual effects, such as a lack of interest in sex, a decreased libido, and a reduced sex drive. LNG-IUS have generally been associated with positive sexual effects, so it could be considered the most innocuous HC regarding sexual side effects. However, more studies are needed because of the inconsistency of current available data.

Finally, with regard to treatment options for sexual dysfunction, few clinical remedies or recommendations exist for women experiencing these sexual side effects [54]. Moreover, no clear guidelines exist for the management of sexual dysfunction potentially associated with CHCs in reproductive-age women [2]. First, when addressing a new sexual complaint, a thorough history using a biopsychosocial approach should be undertaken [18]. A multidisciplinary approach to the management of female sexual dysfunction (FSD) is suggested, particularly when multiple contributing or complicating factors are identified, and this may consist of consultations with other professionals, such as a sex therapist, a pelvic floor physical therapist, and a sexual health specialist [2]. Second, lifestyle counselling should be given by the health providers (Figure 3) [18]. When choosing a new hormonal contraception method, health care providers (HCPs) should give information about all available methods in order to make a shared decision [34]. When CHC-related female sexual dysfunction is suspected, the recommended therapy is the discontinuation of combined hormonal contraceptives with consideration of an alternative method of contraception, such as LNG-IUS, a copper IUC, an etonogestrel implant, the permanent sterilization of either partner when future fertility is not desired, or a contraceptive ring (for women who prefer CHCs for cycle control and non-contraceptive benefits) [2]. The ring appears to be a reasonable alternative to oral CHCs for women with sexual function concerns. Likewise, LARC methods appear to be a reasonable alternative too [2]. Other alternatives could be switching to another combined oral contraceptive [10] or formulations with a shorter hormonal free interval (HFI) [51,61]. Furthermore, with regard to other possible strategies against sexual dysfunction, some studies show positive results on female sexual function with exogenous testosterone [2,18,29], exogenous estrogens [2,6], dehydroepiandrosterone (DHEA) [10,29], tibolone [29], bupropion, and sildenafil [18]. Some alternative options to improve HC-related sexual dysfunction could be vaginal lubricants and moisturizers.

## 5. Conclusions

The results of the studies reviewed seem to indicate that hormonal contraception could influence different aspects of female sexual function. However, there are contradictory results between the different studies regarding the association between sexual dysfunction and hormonal contraceptives, so it could be firmly said that additional research is needed.

Meanwhile, it could be said that hormonal contraception has been associated with different alterations in sexual functioning. So, when addressing a new sexual complaint that is time-related with the beginning of hormonal contraception, health care providers should give information about other methods and try to switch them to a method less associated with sexual dysfunction. Vaginal rings and patches are possible options in case of women preferring combined hormonal contraception who report side effects with the pill.

To conclude, a multidisciplinary approach to the management of female sexual dysfunction is mandatory, and health care providers should give lifestyle counselling apart from proposing different treatment options. An adequate relationship with the patient, as well as the routine monitoring of possible sexual dysfunction, are essential in addressing these difficulties. Undoubtedly, the best contraceptive is one that fulfills the women’s needs with acceptable side effects and agreed with the prescriber.

## Figures and Tables

**Figure 1 jcm-08-00908-f001:**
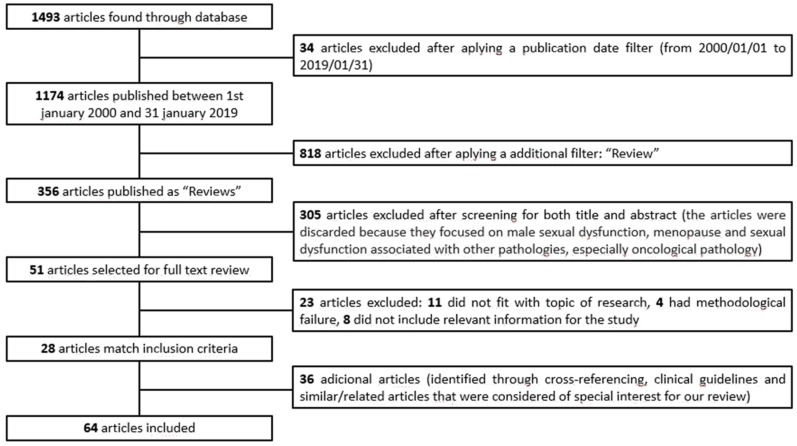
PRISMA flow diagram. (Preferred Reporting Items for Systematic Reviews and Meta-Analyses).

**Figure 2 jcm-08-00908-f002:**
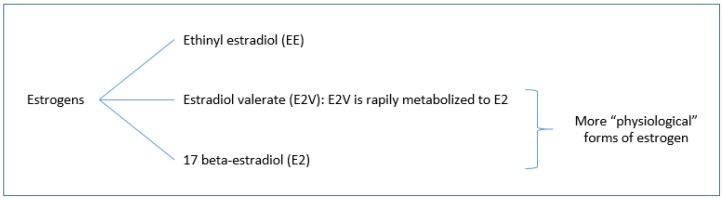
Types of estrogens used in combined oral contraceptives (COCs).

**Figure 3 jcm-08-00908-f003:**
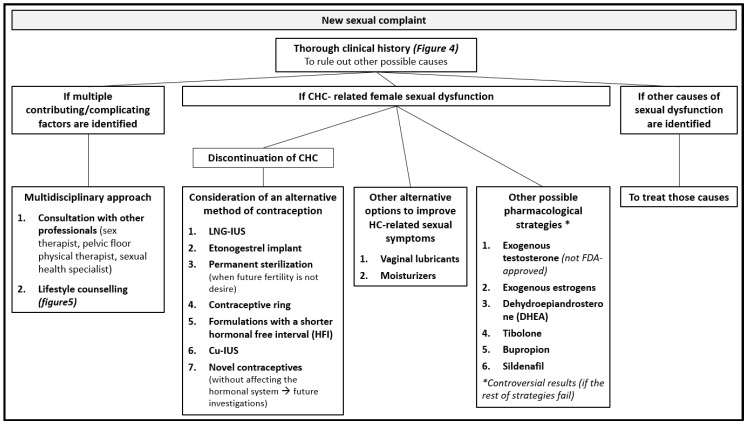
Management strategies for hormonal contraceptive (HC)-related sexual dysfunction.

**Table 1 jcm-08-00908-t001:** Hormonal contraceptives. Route of administration, dosing frequency, mechanism of action, and association with sexual effects.

Hormonal Contraceptives	Route of Administration	Dosing Frequency	Mechanism of Action	Sexual Effects
**Levonorgesetrel-realising intrauterine systems (LNG-IUDs)**	Intrauterine	Inserted by a healthcare provider. Lasts up to 3–5 years, depending on the type. LARC.	Prevention of fertilization: produces a weak foreign body reaction and endometrial decidualization and glandular atrophy changes in the amount and the viscosity of cervical mucus → barrier to sperm penetration Ovulation is likely inhibited in some women but is preserved in most study subjects Endometrial estrogen and progesterone receptors are suppressed	Positive effects. However, more studies are needed
**“The implant”.** **Etonorgestrel implant.**	Subdermal	Inserted by a healthcare provider. Lasts up to 3 years. LARC.	Inhibition of the ovulation and consistently does so until the beginning of the third year of use. Ovarian activity, including estradiol synthesis, is still present. The ENG implant causes thickening of the cervical mucus and changes in the endometrial lining	Negative effects. However, more studies are needed.
**Depot Medroxyprogesterone Acetate (DMPA)**	Intramuscularly	Every three months. SARC/LARC.	Inhibition of the secretion of pituitary gonadotropins → suppressing ovulation Increase of the viscosity of cervical mucus and induction of endometrial atrophy	Mixed results. More studies are needed.
**“The Pill”.** **Combined oral contraceptive**	Oral	Must swallow a pill every day.	Suppression of pituitary gonadotropin secretion → inhibiting ovulation Increase of cervical mucus viscosity → impairing sperm transport Effects on tubal transport → narrowing or eliminating the potential fertilization window Possible endometrial effects Folliculogenesis impairment	Mixed results. More studies are needed.
**“The Mini pill”.** **Progestin-Only Pills (POPs)**	Oral	Must swallow a pill at the same time every day.	Alteration of the cervical mucus: more viscid, less copious → inhibits sperm penetration Possible impairment of sperm motility and decreased tubal cilia activity Negative luteinizing hormone (LH) feedback leads to suppression of ovulation in up to 50% of users	Mixed results. More studies are needed.
**Contraceptive Patch**	Dermal. Is placed on 1 of 4 sites: the buttocks, upper outer arm, lower abdomen, or upper torso, excluding the breast.	Put on a new patch each week for 3 weeks (21 total days). Do not put on a patch during the fourth week.	Similar to the Combined Oral Contraception. Following the first application of the patch, serum hormone levels increase gradually over the first 48 to 72 hours, reach a plateau, and then remain constant during the remainder of the 21-day period. Compared with COCs plasma hormone levels remain constant and the peak levels are lower because first-pass hepatic metabolism and gastrointestinal enzyme degradation are avoided.	Positive effects. Slight increases in sexual function scores were noted with contraceptive patch, but not clinically significant.
**Vaginal Contraceptive Ring**	Vaginal	Put the ring into the vagina yourself. Keep the ring in you r vagina for 3 weeks	Similar to the Combined Oral Contraception Serum hormone levels increase immediately after ring insertion and then decrease slowly over the cycle Gastrointestinal absorption and the hepatic first-pass effect are avoided	Mixed results. More studies are needed.
**Emergency contraceptives**	Route of administration	Dosing frequency		
**Levonorgestrel 1.5 mg**	Oral	Swallow the pills as soon as possible within 3 days after having unprotected sex.		
**Ulipristal Acetate**	Oral	Swallow the pills within 5 days after having unprotected sex.		

**Table 2 jcm-08-00908-t002:** Classification of progestogens used in contraception according to their androgenic potency.

Most Androgenic	Less Androgenic	The Least Androgenic	Antiandrogenic
Norgestrel levonorgestrel	Norethindrone Norethindrone acetate Ethynodiol diacetate	Desogestrel Etonogestrel Norgestimate	Cyproterona acetato Drospirenona Dienogest

**Table 3 jcm-08-00908-t003:** Main data to be collected in the clinical history in case of symptoms of sexual dysfunction.

**Information that should be collected in the medical record by health providers in response to a complaint of sexual dysfunction:**
Current or past psychiatric disorders. Medication use and health problems. History of emotional, physical, or sexual abuse. Beliefs and attitudes regarding sex, menopause, and aging. Body image concerns. Symptoms of depression, anxiety, and sleep problems. Alcohol or drug use and substance use disorders. Health or sexual problems affecting the woman’s sexual partner(s). Relationship discord or communication issues.

**Table 4 jcm-08-00908-t004:** General lifestyle counselling.

Setting aside time to connect with one’s partner Increasing the woman’s exposure to sexual stimuli: erotic literature or films Encouraging maintenance of a healthy weight Ensuring adequate physical activity and sleep Enhancing skills to cope with stress Recommending books women can use for self-education.

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
