# Peer review of "Hormonal Contraceptives, Female Sexual Dysfunction, and Managing Strategies: A Review"

_jcm, 2019, doi:10.3390/jcm8060908_

Round 1
Reviewer 1 Report
The broad topic of this review would be helped by better use of a clearer framework that shows the links between the method, the review's questions
and how the findings are reported.
I suggest that the authors provide more detail of the methods used. Only two databases are listed and it is unclear why these were chosen given the
focus of this review. More detail on how key articles were selected and a summary description of the 64 papers contributing to this review, which could
be described as a scoping review, would be helpful. This description could include a summary of study participants, the countries in which the studies
took place, the types of study designs involved and the range of publication years. This information would give readers a sense of the evidence used
to inform the review.
Care needs to be taken with the use of English in some places as the intended meaning is lost, for example, on page 2 (Line 51) the authors state
"hormonal contraceptives have been used relatively recently", which implies that someone has recently taken these contraceptives, whereas I believe
the intent was to say "the use of hormonal contraceptives is relatively recent".
Not all references appear in the reference list, for example Hall et al 2011 (page 1, line 42) is missing.
Author Response
Author’s reply. Reviewer 1. Round 1. JCM 514811
Dear reviewer.
We thank you very much for the time spent reviewing this article and the kind suggestions for improvement. Below you will find the answers to your
comments and suggestions point by point.
Point 1: I suggest that the authors provide more detail of the methods used. Only two databases are listed and it is unclear why these were chosen
given the focus of this review. More detail on how key articles were selected and a summary description of the 64 papers contributing to this review,
which could be described as a scoping review, would be helpful. This description could include a summary of study participants, the countries in which
the studies took place, the types of study designs involved and the range of publication years. This information would give readers a sense of the
evidence used to inform the review.
This section has been rewritten trying to give more information to the reader as follows:
“The aim of this review is developing, assimilating and synthesizing the existing evidence about the influence of hormonal contraception on female
sexual function. In addition, we intended to identify gaps in knowledge in this field in order to design new studies that may fill those gaps in the future.
Our review focuses on the use of hormonal contraceptives in women of childbearing age and on the influence of these drugs on female sexual function
[1,2]. In addition, the study reviews the differences in the influence of the HCs on the FSF according to the hormonal composition and the mechanism
of action of the different HCs, in order to determine which one has the lowest profile of secondary effects in the sexual area. On the other hand, to our
knowledge, this is the latest effort to offer an overview of the recommended strategies in cases in which the use of HCs is associated with sexual
dysfunction.
To achieve this purpose, we performed a scoping review following PRISMA guidelines (Figure 1). In this review, we selected key articles based on
hormonal contraception and female sexual function. PubMed and Cochrane were chosen as the main databases used due to the extensive contents
of biomedical research they offer, their free access and their ease of use. Our search term combinations were: “hormonal contraception” AND “female
sexual function” OR “female sexual dysfunction”. Filters “Publication date: from 2000/01/01 to 2019/01/31” and “Review” were applied in the search in
order to limit the amount of material available. No language restrictions were applied. Similar and related articles that were considered of special interest
for our review were also included, and they were compiled though cross-referencing. Similarly, some relevant clinical practice guidelines were included.
The 64 papers that were included were chosen because they fit the topic of the review (presenting information about
female sexual dysfunction, hormonal contraception, hormonal variations and their relationship with female sexual function; directly treating the impact
of hormonal contraceptives in female sexual function; or providing relevant information about the management strategies of female sexual dysfunction
associated with the use of HC). We review 6 prospective observational studies, 8 clinical trials, 19 cross-sectional studies, 22 reviews and 9 other works
that include consensus and clinical practice guidelines. Most of the studies were carried out in European countries, although there are also studies
carried out in the US, Asia, Australia and South America. The population of the studies reviewed varies between 40 and 18,787, although in the case
of clinical trials the largest population analyzed is 600 subjects.
We summarized findings and best practice recommendations for addressing a woman’s contraception and its potential association with
sexual function. We excluded those articles that focused on male sexual dysfunction, menopause and sexual dysfunction related to medical
disease, such as oncological pathology. Every attempt was made to combine as much similar data as possible. Institutional review board approval
was not needed for this review.
Point 2: Care needs to be taken with the use of English in some places as the intended meaning is lost, for example, on page 2 (Line 51) the authors
state "hormonal contraceptives have been used relatively recently", which implies that someone has recently taken these contraceptives, whereas I
believe the intent was to say "the use of hormonal contraceptives is relatively recent".
English has been completely revised again by a native medical writer and this sentence has been modified following your suggestion as follows:
“However, the use of hormonal contraceptives is relatively recent.”
Point 3: Not all references appear in the reference list, for example Hall et al 2011 (page 1, line 42) is missing.
This reference has been added and modified to the reference list as follows:
“Hall, K.S.; Trussell, J. Types of combined oral contraceptives used by US women. Contraception 2012, 86, 659–65.”
Reviewer 2 Report
The authors provide a review on female sexual dysfunction and strategies of hormonal contraceptives.
I think the manuscript is well-designed and very informative for researchers as well as clinicians. It is thus almost acceptable in the present form.
However, the authors should revise it according to the following minor concern;
The authors should describe new findings which has not been addressed in the past literatures on the same topic (Ref1& 2), if any.
Author Response
Author’s reply. Reviewer 2. Round 1. JCM 514811
Dear reviewer.
Thank you very much for the time spent reviewing this article and the kind suggestions for its improvement. Below you will find the answers to your
suggestion
Point 1: The authors provide a review on female sexual dysfunction and strategies of hormonal contraceptives. I think the manuscript is well-designed
and very informative for researchers as well as clinicians. It is thus almost acceptable in the present form. However, the authors should revise it
according to the following minor concern; The authors should describe new findings which has not been addressed in the past literatures on the same
topic (Ref1& 2), if any.
Reply:
A new paragraph has been included at the end of the introduction section as follows:
“This review provides a compilation of the existing evidence about the relationship between female sexual function and hormonal contraceptives,
in addition to the existing therapeutic management strategies. This is the first review that includes a summary table, which allows the clinician
to access to the most relevant information at a glance. Likewise, it is the only study that proposes a therapeutic algorithm for the managing
of hormonal contraceptives- related sexual dysfunction”.